# A Pilot Feasibility Study Assessing the Combined Effects of Early Behavioral Intervention and Propranolol on Autism Spectrum Disorder (ASD)

**DOI:** 10.3390/children10101639

**Published:** 2023-09-30

**Authors:** Kathy Hirst, Rachel M. Zamzow, Janine P. Stichter, David Q. Beversdorf

**Affiliations:** 1Thompson Center for Autism and Neurodevelopment, University of Missouri, Columbia, MO 65211, USA; khirst@lakeregional.com (K.H.); stichterj@missouri.edu (J.P.S.); 2Interdisciplinary Neuroscience Program, University of Missouri, Columbia, MO 65211, USA; rachel.zamzow@gmail.com; 3Departments of Radiology, Neurology, and Psychological Sciences, William and Nancy Thompson Endowed Chair in Radiology, University of Missouri, Columbia, MO 65211, USA

**Keywords:** autism, propranolol, early intervention, clinical trial

## Abstract

Autism spectrum disorder (ASD), a neurodevelopmental disorder typified by differences in social communication as well as restricted and repetitive behaviors, is often responsive to early behavioral intervention. However, there is limited information on whether such intervention can be augmented with pharmacological approaches. We conducted a double-blinded, placebo-controlled feasibility trial to examine the effects of the β-adrenergic antagonist propranolol combined with early intensive behavioral intervention (EIBI) for children with ASD. Nine participants with ASD, ages three to ten, undergoing EIBI were enrolled and randomized to a 12-week course of propranolol or placebo. Blinded assessments were conducted at baseline, 6 weeks, and 12 weeks. The primary outcome measures focusing on social interaction were the General Social Outcome Measure-2 (GSOM-2) and Social Responsiveness Scale—Second Edition (SRS-2). Five participants completed the 12-week visit. The sample size was insufficient to evaluate the treatment efficacy. However, side effects were infrequent, and participants were largely able to fully participate in the procedures. Conducting a larger clinical trial to investigate propranolol’s effects on core ASD features within the context of behavioral therapy will be beneficial, as this will advance and individualize combined therapeutic approaches to ASD intervention. This initial study helps to understand feasibility constraints on performing such a study.

## 1. Background

Autism spectrum disorder (ASD) is a neurodevelopmental disorder characterized by social communication differences, and restricted and repetitive behaviors [1]. ASD is known to have a strong genetic component [2], with heritability estimated at 0.83 by the latest, more conservative, analysis [3], and 0.808 in a subsequent five-country cohort [4]. Additionally, epigenetic factors are a critical element in the expression of autism spectrum disorder [5,6]. However, the high degree of genetic heterogeneity makes it difficult to conduct biologically based diagnosis and targeted intervention. In addition, phenotypic heterogeneity [7] and a wide range of behavioral and medical co-occurring conditions, such as attention deficit hyperactivity disorder (ADHD), seizures, and gastrointestinal disorders [8], add to the complexity of treating this prevalent condition.

Maximizing functional independence and improving the quality of life while reducing the impact of core features of this disorder serve as the ultimate goals of treatment [9]. Early diagnosis of ASD leading to behavioral therapy intervention has demonstrated lasting benefits [10,11], in part, due to its influence on behavior when a child’s brain possesses a high degree of plasticity [12]. This critical early stage following diagnosis is an opportune time to integrate pharmacological intervention with behavioral therapy to maximize therapeutic benefits. It has been argued that the integration of pharmacological and behavioral intervention may improve outcomes for individuals with ASD [13]. However, this combined therapeutic approach has not been well studied in the context of this population [14]. And while pharmacotherapy is often a significant component of ASD treatment, it is most often directed at co-occurring psychiatric traits which include agitation and obsessive behaviors [9,15,16]. Some reports have identified benefits of oxytocin for social cognition in ASD [17,18], while others have explored glutamate antagonists for their effects on social and language functioning [19,20]. One study demonstrated social benefits of the GABA agonist arbaclofen [21]; however, the drug did not succeed in subsequent clinical trials [22]. Phytochemical sulforaphane has also shown improvements in social responsiveness and behavioral disturbances [23]. However, no pharmacological agents have yet been shown to improve the core features of ASD in larger double-blinded, placebo-controlled trials.

There is existing evidence which supports autonomic dysregulation in individuals with ASD [24,25,26]. Hence, targeting the noradrenergic system may be advantageous in the treatment of individuals with ASD. Among the noradrenergic agents, emerging literature has suggested a potential benefit from the nonselective β-adrenergic antagonist propranolol. A report highlighted that individuals with ASD treated using β-adrenergic antagonists, such as propranolol, demonstrated improved language and social behaviors in an open label case series study [27]. Propranolol, a centrally and peripherally active β -adrenergic antagonist, dampens sympathetic nervous system activity and reduces adrenergic signaling by competitively and non-selectively blocking both β1- and β2-adrenergic receptors. In addition to its common use for hypertension with minimal adverse effects, propranolol also has anxiolytic properties, which have led to prominent off-label use for test [28] and performance anxiety [29,30]. This agent has since been explored for its effects on symptoms of post-traumatic stress disorder [31,32] and acute cocaine withdrawal [33,34]. Studies in individuals without neurodevelopmental diagnoses found greater performance on a verbal problem-solving task with single-dose administration of propranolol than with other agents [35,36], and found that propranolol reversed the impairments in problem-solving induced by psychosocial stressors [37]. Further investigation revealed that the benefits of propranolol, as compared with placebo, are more apparent when task difficulty is increased [38].

As mentioned above, the first investigation of the effects of propranolol in the ASD population demonstrated improvements in language and sociability in an uncontrolled case series [27]. Subsequent single-dose psychopharmacological challenge studies found improvements in verbal abilities [39,40], working memory [41], and facial scanning [42]. A more recent study showed that performance on a social task involving a naturalistic conversation was significantly greater in adults and adolescents with ASD treated with propranolol, as compared to placebo, in a within-subjects design [43]. These findings, coupled with a recent literature review, suggest that social abilities as well as emotional, behavioral and autonomic dysregulation may be improved by propranolol in individuals with ASD [44]. Additionally, there are recently reported benefits for propranolol on behavior [45] and anxiety [46].

Assessing the effects of propranolol within the context of behavioral therapy would be valuable for advancing combined therapeutic approaches to ASD intervention. Early intensive behavioral intervention (EIBI) is a comprehensive individualized approach to intervention based on the principles of Applied Behavior Analysis (ABA) [47]. EIBI is designed to be rigorous, with 20 to 30 hours (h) of therapy sessions per week for multiple years, with most children beginning therapy at around three to four years of age [48]. With goals of teaching adaptive functioning and minimizing challenging behavior, board certified behavior analysts (BCBAs) break individual skills into small components and teach them hierarchically. Consistent feedback and rewards are used as prompts to maximize success. This intervention also involves data collection methods to monitor progress and refine treatment goals. An initial study of EIBI’s efficacy found that children with ASD demonstrated improvements in IQ and educational functioning following an EIBI protocol of at least 40 h of therapy per week for a period of two or more years [49]. A follow-up examination of the same children demonstrated that these improvements were maintained in adolescence [50]. Subsequent studies have shown significant improvements in IQ, language development, educational skills, and adaptive behaviors following EIBI [51,52,53,54,55,56,57,58,59,60]. However, only a few studies have specifically examined the changes in social functioning following EIBI [56,57,61]. Consequently, it is important to explore how the effects of EIBI on this core feature can be augmented with pharmacological intervention, particularly via the introduction of propranolol.

Previous examinations on propranolol effects on ASD traits were conducted in adult and adolescent populations. Given the advantages of early intervention in ASD, it is crucial to investigate the impact of this agent in younger individuals. Additionally, this agent is yet to be explored for benefits in ASD in a serial-dose setting or in a combined therapeutic context alongside behavioral intervention. Propranolol has been previously utilized in children to treat pediatric migraine [62,63,64] and infantile hemangioma [65], demonstrating its safety for this population. Therefore, an investigation of propranolol’s effects in a trial with concurrent EIBI in children with ASD would enhance understanding of this agent’s potential benefit for this population. Accordingly, we performed a 12-week, double-blind, placebo-controlled parallel study to examine the combined therapeutic effects of propranolol and EIBI on social interaction in children with ASD. Our ultimate hypothesis is that with recently reported benefits for propranolol on behavior [45] and anxiety [46], improving behaviors and anxiety during EIBI might yield benefits in the social domain, making this agent particularly interesting as an add-on treatment in this young population given its long history of excellent tolerability in pediatric populations [62,63,64,65]. This specific study serves as a feasibility study for subsequent exploration of this hypothesis.

## 2. Methods

Nine individuals with ASD, ages three to ten, were enrolled in the double-blind pilot trial of propranolol coupled with EIBI (NCT02428205). The trial was conducted at a single site and consisted of a 12-week treatment period. The participants, care providers, investigators and outcomes assessors were each blinded to the subject treatment group. None of the participants had any comorbid conditions.

ASD diagnosis of participants was confirmed if they met or exceeded the clinical cutoff for ASD on the Autism Diagnostic Observational Schedule-2 (ADOS-2) [66]. All participants were recruited from the Thompson Center for Autism and Neurodevelopment at the University of Missouri, Columbia, Missouri, wherein they were undergoing an EIBI program or a program of similar intensity, following the ABA principles. Given the highly individualized nature of EIBI, each participant’s therapy differed in frequency of sessions per week and length, as well as defined behavioral targets and the methods used to reach them. EIBI fidelity data and information regarding behavioral targets were collected from BCBA and the implementers working with each participant. All participants continued the EIBI program throughout the duration of their participation in this trial.

Participants with risk factors for exposure to propranolol, including current diagnoses of asthma, bradycardia, diabetes, thyroid disease, depression, or use of any noradrenergic agents, were excluded. Those with potentially confounding diagnoses, such as major head trauma or neurological or psychiatric diagnoses, were also excluded. Lastly, children with a heart rate of less than 60 or a systolic blood pressure of less than 75 at the outset of the study were excluded. All procedures were conducted in accordance with the approval of the Institutional Review Board (IRB) of the University of Missouri.

EIBI: The EIBI program provides long-term services for individuals with autism spectrum disorder by teaching them new skills and minimizing or eliminating their challenging behaviors. The therapeutic approach used in the EIBI service is based on the principles of Applied Behavior Analysis (ABA). In general, the program first teaches skills in a reduced setting, breaking skills down to simplest components, and then teaches skills hierarchically to master them, while giving consistent feedback, using rewards to motivate and building new skills, providing prompts to maximize success, and collecting data to document progress on skill acquisition and problematic behavior reduction.

Interventions are comprehensive and individualized based on the developmental level of the individual. Research on treatment effectiveness was first conducted by Lovaas in 1987 [49], which indicated that 90% of children substantially improved after participating in EIBI services compared to the control group. Replications of the study have been carried out and have yielded similar results in randomized controlled trials of EIBI services, and results showed significant differences in IQ, language development, and academic skills compared to the control group. [52] demonstrated that a focused, individualized EIBI program is far superior to an eclectic special education approach that uses a variety of treatments. Sallows et al. [57] found that after participating in individualized EIBI services, 48% of children showed rapid learning, achieved average posttreatment scores, and at age 7, were succeeding in regular education classrooms. Cohen et al. [51] showed that children receiving such EIBI services scored significantly higher in IQ and adaptive behavior scores than the comparison group. Eldevik et al. [54] also demonstrated that the outcomes for the EIBI group were significantly better than those for the control and comparison groups. An EIBI program that utilizes a developmentally individualized approach demonstrated to be most effective in these studies. Thus, with this trial, the ultimate goal is to determine whether concomitant administration of propranolol further enhances the gains achieved through the EIBI program, such as the anxiolytic effects and the effects on social interaction, as compared to placebo group. While interventions within the EIBI directly target communication, the tasks used for assessment in this trial are distinct from those used in the EIBI program. The staff members administering the EIBI program were blinded to the treatment group, as was the rest of the team.

Outcome measures: Outcome measures included assessments of social functioning, language, anxiety, adaptive behaviors, and other ASD-related behaviors using several tools: The Vineland Adaptive Behavior Scale—Second Edition (VABS) is a validated assessment used for the range of our participants’ ages, which yields standard scores in Communication, Daily Living Skills, Socialization, and Motor Skills [67]. The Aberrant Behavior Checklist (ABC) is a widely used assessment tool for interventions in a range of cognitive disorders. It is a 58-item questionnaire for the parent/caregiver, rated on a four-point scale (0 = not at all a problem, 3 = the problem is severe). Items are scored on five subscales: I: Irritability, Agitation, Crying; II: Lethargy, Social Withdrawal; III: Stereotypic Behavior; IV: Hyperactivity, Noncompliance; V: Inappropriate Speech. Each subscale was utilized in our assessment. It has been validated and utilized in a wide range of ages and cognitive conditions including ASD [68]. The General Social Outcome Measure (GSOM) [69] has been previously used to demonstrate the effects of propranolol on single-dose psychopharmacology studies [43]. The Social Responsiveness Scale—Second Edition (SRS-2) is a widely used and validated 65-item rating scale measuring deficits in social behavior associated with ASD, completed by raters who have at least one month of experience with the rated individuals [70,71]. The Preschool Language Scale—Fifth Edition (PLS-5), an individually administered, norm-referenced, validated play-based instrument assesses developmental language skills in children from birth to 7 years and 11 months, providing standard scores, growth scores, language age equivalents, and percentile rank scores. It has been found useful for tracking progress across time. The instrument includes manipulatives for use in test administration. The areas assessed are as follows: Attention; Play; Gesture; Vocal Development; Social Communication; Semantics; Language Structure; Integrative Language Skills; Emergent Literacy Skills [72]. The Preschool Anxiety Scale (PAS), a validated 28-item scale that is completed by a parent/guardian, assesses anxiety in children between the ages of 2 ½ and 6 ½ years old. The 28 items provide an overall measure of anxiety, in addition to scores on five subscales assessing a specific aspect of child anxiety: 1: Generalized Anxiety, 2: Social Anxiety, 3: Obsessive Compulsive Disorder, 4: Physical Injury Fears and 5: Separation Anxiety. This assessment is intended to provide an indicator of the number and severity of anxiety symptoms experienced by younger children [73]. The Autism Impact Measure (AIM) is a validated tool developed to assess the impact of the ASD-associated behaviors. Participants are asked a series of 41 questions regarding the frequency and the impact or interference resulting from a series of ASD-associated behaviors in order to determine effects, and includes subdomains on peer interaction, social reciprocity, atypical behavior, communication, and repetitive behavior [74]. The Sensory Experiences Questionnaire (SEQ) is a validated caregiver report assessment intended to be used by researchers and clinicians to characterize the sensory features in children ages 2–12 years with autism spectrum disorder (ASD) and/or developmental disabilities (DD) in social and nonsocial contexts [75]. All measurements and assessments were performed for a total of three times during separate testing sessions throughout the study: (1) prior to drug administration, for establishment of a baseline (week 0); (2) at week 6; (3) at the end of week 12. GSOM and SRS-2 were used as primary outcomes given the results of previous studies of propranolol in ASD [43]. All team members were trained in the administration of these measures.

Following the successful validation of the eligibility criteria and after receiving the informed consent, participants were randomized to begin the oral dosing of propranolol or placebo one hour prior to each scheduled EIBI session. Randomization was performed with a randomization table generated by the pharmacy. Drug was administered by the participant’s parent/caregiver or a medical professional (e.g., school nurse). Dosing frequency varied by subject depending on their therapy schedule. Dosages were adjusted according to body weight from the minimum dose (10 mg) of propranolol used safely in otherwise healthy adults for testing anxiety (Table 1), and administered one hour prior to each therapy session. Potential participants under 15 kg were excluded.

Bivariate descriptive statistics were conducted to summarize central tendency and variation in assessments at each time point (i.e., baseline, 6-week, and 12-week) within each treatment arm. While the sample was insufficient to allow for the analysis for subjects completing the study, we conducted a conservative Last Observation Carried Forward analysis to allow for a preliminary comparison between groups.

## 3. Results

Nine participants were enrolled, of whom six completed the six-week assessment (four in the propranolol group), and five completed the twelve-week assessment (three in the propranolol group) of the double-blind study. One participant in the propranolol group dropped and two individuals in the placebo group dropped out after the first visit. The randomized sample was 100% male, and had a mean age of 5.87 years (range 3–10 years, ±2.41 standard deviation) (see Table 2 for full enrollment demographics; see Figure 1 for flow diagram). All participants averaged three EIBI sessions per week (seven subjects had three sessions per week, one subject had two sessions per week, and one subject had four sessions per week), resulting in equitable opportunities to meet the treatment goals across the entire sample size. Four caregivers withdrew consent during the study due to observing no change in the participant (*n* = 1, propranolol arm), worsening behaviors of the participant (*n* = 1, placebo arm), and unknown reasons (*n* = 2, propranolol and placebo arms). Overall, four adverse events were noted during the study: aggression, irritability, insomnia and fatigue. Only one of these events was recorded during the propranolol treatment (fatigue), while the other three were noted during the placebo treatment. No subjects discontinued propranolol treatment due to any treatment-related adverse effects.

While the number of participants completing the study for each group was too small for any quantitative analysis, for subjects who completed the study, the total GSOM score in the propranolol group improved from 31.7 (±16.9 standard deviation (SD)) to 48.3 (±14.0 SD) from baseline to week 12; the score of the placebo group was 20.5 (±6.4 SD) at baseline and 20.0 (±15.6 SD) at week 12. Neither group had an observed change in the SRS, as the propranolol group’s score was 76.0 (±12.0 SD) at baseline and 79.3 (±17.0 SD) at week 12; the placebo group’s score was 75.0 (±1.4 SD) at baseline and 72.0 (±2.8 SD) at week 12. Similarly, there were no changes in the mean scores for the Vineland, AIM, PLS, and SEQ assessments, expect for an isolated observation in the Peer Interaction Subdomain Impact wherein the raw total mean changed from 31.3 (±5.0 SD) at baseline to 23.7 (±8.7 SD) at 12 weeks with propranolol, while the values for the placebo group was 26.5 (±9.2 SD) at baseline and 24.5 (±6.4 SD) at 12 weeks (see Table 3). The high degree of variability in the PAS-5 and the ABC scores limited any meaningful description of their results, as the standard deviations were nearly as large as the means in each case. Other measures attempted are not included as many participants refused to complete the tasks. Any other differences apparent above should be interpreted with utmost caution as the small sample size did not allow for significant statistical comparison. Though the Last Observation Carried Forward (LOCF) analysis was performed, due to the small sample size and the number of zero change in the follow-up measures (by the nature of LOCF for early dropouts), no statistical differences were observed in any of the measures at any time point. Only some trends were found for improved peer interaction with a couple of domains in the AIM, and also increased issues with sensory experiences, which had a worsening trend (see Appendix A). However, none of these results withstand correction for multiple measures.

## 4. Discussion

ASD is a heterogeneous neurodevelopmental disorder with relatively few comprehensive and combined therapeutic treatment options that target the core features. Hence, we proposed to perform a feasibility study for future studies to examine the effects of propranolol on social interaction in children with ASD undergoing EIBI or a program of similar intensity, following the ABA principles, in a 12-week double-blinded, placebo-controlled parallel study. The ultimate goal was to determine whether concomitant administration of propranolol would augment EIBI’s effect to enhance clinical benefit by improving social functioning, language, anxiety, adaptive behaviors, and other ASD-related behaviors. Only a few studies to date have explored how pharmacological intervention affects behavioral therapy outcomes, and agent exploration in younger ASD patients continues to be minimal [76]. Propranolol was of particular interest due to its long established effects on situational anxiety [29,30], its history of being well tolerated in young children [62,63,64,65], and past evidence suggesting its effects in older children with ASD [45,46], wherein it is believed to target the access to neural networks and the sympathetic system, regardless of the degree of autonomic dysfunction (see Beversdorf [77] for review).

Though designed as a placebo-controlled parallel arm study, this study was underpowered to detect potential clinical changes. Subject variation was extensive in terms of EIBI behavioral targets and cognition with limited baseline functional capacity, leading to varying levels of tolerance and assessment comprehension among the small project sample. These facets resulted in numerous refusals of the subjects to complete the assessments, as well as limited behavioral assessment data for the subjects with lower cognitive states. From an ethical standpoint, we could not control or modify the EIBI behavioral target plan for the participants, which were selected for the best care and treatment of the patients. For these reasons, there was insufficient data to make any determination as to whether particular subject clinical characteristics may have predicted positive clinical outcomes. Any subsequent study would need to incorporate a sample sufficient enough to account for the inherent variability in the EIBI or related programs.

Overall, propranolol was tolerated well. Only one subject reported fatigue during propranolol treatment. Fatigue is a known potential side effect of propranolol treatment; this was expected, and it did not result in cessation of the drug treatment. This reinforces the fact that propranolol continues to have a well-established safety profile and is tolerable in a pediatric population. Findings from the current study provide direction for further exploration of propranolol as an effective treatment for ASD symptoms, and inform future combined therapeutic trials for optimal outcome measures. Behavioral therapy, particularly with an intensive and individualized approach, is a foundational treatment for ASD. Hence, potential future work would need to account for the differences in targeted therapy. In addition, explorations of potential treatment response markers, including anxiety and autonomic nervous system functioning, would allow for the future development of individualized treatment options that are likely to be more effective. A large population would need to be recruited to have a sufficient number of participants who are able to successfully complete the trial to fully assess the effects of this combined therapeutic approach, with assessments designed for optimal targeting of baseline performance levels to avoid floor effects. But the present study suggests that most participants would tolerate the procedures and medications well. Recent studies have demonstrated the effects of propranolol on severe behaviors in individuals with ASD [45], as well as a new double blind placebo controlled trial of propranolol in ASD suggesting an effect on anxiety in ASD [46]. However, these studies and most others in the literature have targeted older children and adults. This raises the question as to whether early intervention with this agent might be a beneficial supplement to behavioral interventions in young children in a subsequent clinical trial, accounting for the array of hurdles revealed in this feasibility study.

## Figures and Tables

**Figure 1 children-10-01639-f001:**
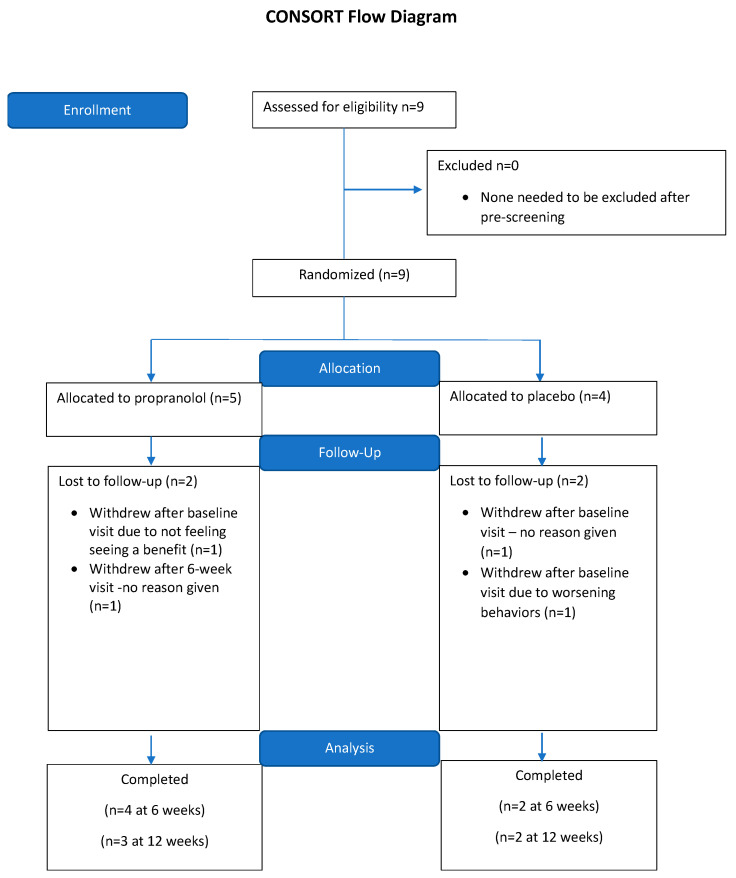
Consort diagram for study enrollment.

**Table 1 children-10-01639-t001:** Body weight-adjusted propranolol doses.

Bodyweight (kg)	Dose (mg)
>30	4
22.5–30	3
15–22.5	2
<15	N/A

**Table 2 children-10-01639-t002:** Demographic data of the study population.

	Number of Patients Propranolol Group	Number of Patients Placebo Group
Gender		
Male	5	4
Female	0	0
Age (years)		
3–6	4	2
7–10	1	2
Body weight (kg)-adjusted doses (mg)		
>30 kg, 4 mg	2	2
22.5–30 kg, 3 mg	1	1
15–22.4 kg, 2 mg	2	1

**Table 3 children-10-01639-t003:** Summary of outcomes for participants completing the study (wk = week).

Task	Propranolol: Baseline	Propranolol: 12 wk	Placebo: Baseline	Placebo: 12 wk
GSOM total	31.7 (±16.9 SD)	48.3 (±14.0 SD)	20.5 (±6.4 SD)	20.0 (±15.6 SD)
SRS	76.0 (±12.0 SD)	79.3 (±17.0 SD)	75.0 (±1.4 SD)	72.0 (±2.8 SD)
PAS	78.0 (±24.3 SD)	87.7 (±32.9 SD)	64.0 (±19.8 SD)	71.5 (±30.4 SD)
SEQ	91.0 (±26.0 SD)	102.3 (±30.7 SD)	116.0 (±29.7 SD)	102.0 (±35.4 SD)
Vineland: communication	81.7 (±12.1 SD)	76.0 (±26.2 SD)	75.0 (±22.6 SD)	74.5 (±29.0 SD)
Vineland: daily living	81.3 (±6.4 SD)	69.7 (±20.4 SD)	76.5 (±14.8 SD)	77.0 (±22.6 SD)
Vineland: social	70.3 (±8.5 SD)	70.7 (±10.2 SD)	66.0 (±26.9 SD)	66.5 (±23.3 SD)
AIM: peer interaction Subdomain impact raw score	31.3 (±5.0 SD)	23.7 (±8.7 SD)	26.5 (±9.2 SD)	24.5 (±6.4 SD)
AIM: social reciprocity Subdomain impact raw score	30.7 (±1.5 SD)	31.3 (±3.1 SD)	37.5 (±9.2 SD)	32.0 (±9.9 SD)
AIM: atypical behavior Subdomain impact raw score	31.3 (±11.0 SD)	36.7 (±15.3 SD)	26.0 (±5.7 SD)	25.0 (±1.4 SD)
AIM: communication Subdomain impact raw score	39.0 (±6.0 SD)	36.0 (±11.8 SD)	45.0 (±8.5 SD)	41.0 (±8.5 SD)
AIM: repetitive behavior Subdomain impact raw score	38.7 (±16.2 SD)	41.7 (±19.7 SD)	49.5 (±7.8 SD)	45.0 (±15.6 SD)

## Data Availability

Data will be made available in a de-identified manner upon request. Raw data will not be publicly available for privacy reasons.

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
