# Peer review of "A Pilot Feasibility Study Assessing the Combined Effects of Early Behavioral Intervention and Propranolol on Autism Spectrum Disorder (ASD)"

_children, 2023, doi:10.3390/children10101639_

Round 1

Reviewer 1 Report (New Reviewer)

The introduction lacks a concept of why this particular medication is the subject of the study and not another.

The description of the sample would be enriched if more information was given about the results of the tests that were performed on the participants and information about comorbidity, biological test data, duration and type of therapy received.

I would recommend to present the results more clearly and in more detail.

The discussion repeats the results of the study without offering an idea of the mechanism of the effect and change, even if that idea can not be fully proved, but it can be used to support the need for future research in the field.

Author Response

The introduction lacks a concept of why this particular medication is the subject of the study and not another.

We have added further rationale as to why propranolol is examined. The rationale was in paragraphs 3 and 4, and we have added to this, and wrapped back to this as why it is an add-on in this population.

The description of the sample would be enriched if more information was given about the results of the tests that were performed on the participants and information about comorbidity, biological test data, duration and type of therapy received.

We included the comorbid conditions- there were none- these were small otherwise healthy children- and the duration of therapy.  All of the treatment results are included.

I would recommend to present the results more clearly and in more detail.

We have added a supplementary table that includes last observation carried forward for greater detail, allowing a conservative analysis.

The discussion repeats the results of the study without offering an idea of the mechanism of the effect and change, even if that idea can not be fully proved, but it can be used to support the need for future research in the field.

We don’t want to overstate, but do now add mechanism and what we do observe, stated very cautiously.

Reviewer 2 Report (New Reviewer)

Goodtime,

Thank you for your resaerch, Ihave reviewed , and some questions remain,

What is the differences between your study and"

The Safety and Effectiveness of High-Dose Propranolol as a Treatment for Challenging Behaviors in Individuals With Autism Spectrum Disorders

the protocol or program is unclear

Author Response

Thank you for your research, I have reviewed , and some questions remain,

What is the differences between your study and"

The Safety and Effectiveness of High-Dose Propranolol as a Treatment for Challenging Behaviors in Individuals With Autism Spectrum Disorders

We have clarified how our study differs.

the protocol or program is unclear

We have added details on the protocol, clarifying that all participants continued through the EIBI throughout the trial, and added a consort figure.

Reviewer 3 Report (New Reviewer)

This pilot study explores how the integration of propronolol and behavioral intervention may improve social interaction in ASD. Due to the relatively low participant pool, refusals of the subjects to complete assessments, and limited behavioral assessment data for the subjects with lower cognitive states, there was insufficient power to make any determination. Additionally, there are several methodological concerns and deficiencies in the presentation of data, that need to be addressed. Other sections such as introduction and discussion are nicely narrated.

Major issues:

In Methods

It is not mentioned how many were allocated in the placebo group.

How was the randomization performed?

Statistical methods (including the tests and software) are not described.

Line 201 “Peer Interaction Subdomain Impact” discussed in the results is not mentioned in the methodology section.

How was the assessment fidelity established and maintained?

Include a flow diagram of participants through the trial.

https://www.nature.com/articles/mp2016168/figures/1

Results.

Tablate the demographic data of the study population separately for the treatment group and placebo group.

Statistical findings are underreported.

Tabulate the outcome measures (with subdomains); baseline,  12-week assessment, estimated effect, p-value, etc.

 https://www.nature.com/articles/mp2016168/tables/4

Minor comments

In the discussion

Line 216; “The ultimate goal will be to determine” should be written in past tense, isn't it?

In the introduction summarise evidence  for autonomic dysfunction (sympathetic overactivity?) in ASD, and link that science to the mechanism of action of propranolol.

There is a duplication of references.

48 & 49. Myers, S. M., & Johnson, C. P. (2007). Management of children with autism spectrum disorders. Pediatrics, 120, 1162-1182.

There are 12  self-citations out of a total of 73. Some self-citation can be avoided with alternative references.

ẹg. 7. Beversdorf, D. (2008). Therapeutic interventions in autism: A review for primary care physicians. Mo Med, 105, 390-395.

The citation at directed at co-occurring psychiatric traits which include agitation and obsessive behaviors (Beversdorf...." also looks inappropriate. Because the paper focuses on the effect of propranolol on verbal problem-solving, agitation, or obsessive behavior.

Author Response

This pilot study explores how the integration of propronolol and behavioral intervention may improve social interaction in ASD. Due to the relatively low participant pool, refusals of the subjects to complete assessments, and limited behavioral assessment data for the subjects with lower cognitive states, there was insufficient power to make any determination. Additionally, there are several methodological concerns and deficiencies in the presentation of data, that need to be addressed. Other sections such as introduction and discussion are nicely narrated.

Major issues:

In Methods

It is not mentioned how many were allocated in the placebo group.

How was the randomization performed?

We have added this information.

Statistical methods (including the tests and software) are not described.

The sample was too small for valid statistical comparison, as we clarify.  However, we do now mention a new Last Observation Carried Forward analysis.

Line 201 “Peer Interaction Subdomain Impact” discussed in the results is not mentioned in the methodology section.

We have added the subdomains in the methods for clarity.

How was the assessment fidelity established and maintained?

We have clarified the training of the team                                                                                               .

Include a flow diagram of participants through the trial.

https://www.nature.com/articles/mp2016168/figures/1

We have added a flow diagram.

Results.

Tablate the demographic data of the study population separately for the treatment group and placebo group.

This makes sense- we have added this.

Statistical findings are underreported.

Tabulate the outcome measures (with subdomains); baseline,  12-week assessment, estimated effect, p-value, etc.

See above, the sample is too small for statistical comparison, but have tabulated the outcomes otherwise.  We did add a Last Observation Carried Forward supplementary table to allow a preliminary comparison.

 https://www.nature.com/articles/mp2016168/tables/4

Minor comments

In the discussion

Line 216; “The ultimate goal will be to determine” should be written in past tense, isn't it?

Correct- we have fixed this.

In the introduction summarise evidence  for autonomic dysfunction (sympathetic overactivity?) in ASD, and link that science to the mechanism of action of propranolol.

We have added this to the end of the first paragraph of the Discussion- thanks!

There is a duplication of references.

48 & 49. Myers, S. M., & Johnson, C. P. (2007). Management of children with autism spectrum disorders. Pediatrics, 120, 1162-1182.

We have fixed this- thanks for catching it.

There are 12  self-citations out of a total of 73. Some self-citation can be avoided with alternative references.

ẹg. 7. Beversdorf, D. (2008). Therapeutic interventions in autism: A review for primary care physicians. Mo Med, 105, 390-395.

We are the main research group publishing on propranolol in autism, and were simply trying to share the literature.  The proposed 2008 reference precedes nearly all of these other papers, and would not serve as an appropriate alternative reference.  Instead, we have added the only other propranolol and autism references from other groups for completeness.

The citation at “directed at co-occurring psychiatric traits which include agitation and obsessive behaviors (Beversdorf...." also looks inappropriate. Because the paper focuses on the effect of propranolol on verbal problem-solving, agitation, or obsessive behavior.

No- verbal problem solving paper is Beversdorf et al 2008.  The Beversdorf 2008 reference is a review on treatment of ASD.  That reference is correct.

Round 2

Reviewer 2 Report (New Reviewer)

Good time,

I have reviewed it again, but the responses to the questions remain unclear.

regard

Reviewer 3 Report (New Reviewer)

Dear Author,

Congratulations! Many thanks for the point-by-point response. The changes done are aggreable. 

I see extensive descriptions of EIBI and outcome measures have been incorporated. I would make the methodology section a little more concise for a Brief Report, but it's entirely your decision.  

All the best

This manuscript is a resubmission of an earlier submission. The following is a list of the peer review reports and author responses from that submission.

Round 1

Reviewer 1 Report

- There are several main issues in this brief research article:

- References related to ASD in the Introduction section are too old. Approximately 50% of genetic heritability for ASD should be updated, it seems too high. No mention of epigenetic involvement has be done.

- The enrolled groups are too small (also as stated by authors), but it is not clear how many children were enrolled in the treatment-group and how many in the placebo-group. Was just one child in this last group?

- However, the study design seems to be poor. Authors wrote: "Subject variation was wide in terms of EIBI behavioral targets and cognition with limited baseline functional capacity, leading to varying levels of tolerance and assessment comprehension among the small project sample". 

This means that authors ab initio recognized that a baseline was hard to reach. In other term, if the EIBI is widely different among the children, how is possible to normalize results after propranolol administration? It appears to be difficult also with a large number of enrolled subjects.

- On this topic, see also: London EB, Yoo JH, Fethke ED, Zimmerman-Bier B. The Safety and Effectiveness of High-Dose Propranolol as a Treatment for Challenging Behaviors in Individuals With Autism Spectrum Disorders. J Clin Psychopharmacol. 2020 Mar/Apr;40(2):122-129. doi: 10.1097/JCP.0000000000001175. PMID: 32134849.

- Finally, I do not agree with the authors. The current findings of this study should discourage large clinical trials or using this drug for ASD management.

Author Response

- There are several main issues in this brief research article:

- References related to ASD in the Introduction section are too old. Approximately 50% of genetic heritability for ASD should be updated, it seems too high. No mention of epigenetic involvement has be done.

We did not update the text from the original proposal in this section.  We have updated this now.

- The enrolled groups are too small (also as stated by authors), but it is not clear how many children were enrolled in the treatment-group and how many in the placebo-group. Was just one child in this last group?

We have clarified the number of participants in each group.  There were 2 in the placebo group.  However, from a feasibility standpoint, critically, more were in the propranolol group.

- However, the study design seems to be poor. Authors wrote: "Subject variation was wide in terms of EIBI behavioral targets and cognition with limited baseline functional capacity, leading to varying levels of tolerance and assessment comprehension among the small project sample". 

This means that authors ab initio recognized that a baseline was hard to reach. In other term, if the EIBI is widely different among the children, how is possible to normalize results after propranolol administration? It appears to be difficult also with a large number of enrolled subjects.

The EIBI is personalized to treat individual behavioral issues with each child.  It would not be ethical to modify this treatment plan just to add on a drug.  Future studies would need a sufficiently large sample to statistically account for this variation.  The purpose of this, though was feasibility.  We have now discussed this more thoroughly.

- On this topic, see also: London EB, Yoo JH, Fethke ED, Zimmerman-Bier B. The Safety and Effectiveness of High-Dose Propranolol as a Treatment for Challenging Behaviors in Individuals With Autism Spectrum Disorders. J Clin Psychopharmacol. 2020 Mar/Apr;40(2):122-129. doi: 10.1097/JCP.0000000000001175. PMID: 32134849.

We have added this important reference.

- Finally, I do not agree with the authors. The current findings of this study should discourage large clinical trials or using this drug for ASD management.

We have revised our conclusion, but also add that there is now a double blinded placebo controlled trial for propranolol which does lend support for further pursuit of this question, in addition to the London et al paper.  These are discussed for at least asking the question in future study.

Reviewer 2 Report

The need for research is well described in the introduction, but it needs to be described with reference to recent literature.

Please provide a specific research question or hypothesis.

Evidence of validity and reliability of tools used for outcome measurement should be found in the existing literature and presented. Evidence of validity and reliability of tools used for performance measurement should be found in the existing literature and presented.

Behavioral interventions should be described in detail.

Even with a small number of samples, the results should be tabulated in an easy-to-understand format.

It is also very likely that the dropout rate was so high that it affected the validity of the results of this study. In the discussion part, the significance of the study and the need for the further study should be supplemented, and this study should also be tried again to generalize the research results based on these supplementary points.

Author Response

The need for research is well described in the introduction, but it needs to be described with reference to recent literature.

Please provide a specific research question or hypothesis.

The background was not sufficiently updated from the original proposal.  We have updated this now, and have clarified our hypothesis.

Evidence of validity and reliability of tools used for outcome measurement should be found in the existing literature and presented. Evidence of validity and reliability of tools used for performance measurement should be found in the existing literature and presented.

We have now added these for the assessment tools utilized, as requested.

Behavioral interventions should be described in detail.

We have now expanded the description of the EIBI as requested.

Even with a small number of samples, the results should be tabulated in an easy-to-understand format.

This is an excellent point- we have added a table of these results for clarity.

It is also very likely that the dropout rate was so high that it affected the validity of the results of this study. In the discussion part, the significance of the study and the need for the further study should be supplemented, and this study should also be tried again to generalize the research results based on these supplementary points.

We agree, which is why the focus of this small report is as a feasibility study, and yes this should be replicated with supplementary study designed around the understanding of the challenges in retention discovered herein, as we now discuss.